# Transcranial Magnetic Stimulation (rTMS) on the Precuneus in Alzheimer’s Disease: A Literature Review

**DOI:** 10.3390/brainsci13091332

**Published:** 2023-09-15

**Authors:** Bruno Millet, Stéphane Mouchabac, Gabriel Robert, Redwan Maatoug, Thibaut Dondaine, Florian Ferreri, Alexis Bourla

**Affiliations:** 1Service de Psychiatrie Adulte de la Pitié-Salpêtrière, Institut du Cerveau, ICM, Sorbonne Université, AP-HP, 75013 Paris, France; b.millet@aphp.fr (B.M.);; 2ICRIN Psychiatry (Infrastructure of Clinical Research in Neurosciences-Psychiatry), Brain Institute (ICM), INSERM, CNRS, 75013 Paris, France; stephane.mouchabac@aphp.fr (S.M.); florian.ferreri@aphp.fr (F.F.); 3Department of Psychiatry, Hôpital Saint-Antoine, Sorbonne Université, AP-HP, 75012 Paris, France; 4Pôle Hospitalo-Universitaire de Psychiatrie Adulte, Centre Hospitalier Guillaume Régnier, 35000 Rennes, France; 5U1228 Empenn, UMR 6074 IRISA, Campus de Beaulieu, 35042 Rennes, France; 6Neuroscience et Cognition, Degenerative and Vascular Cognitive Disorders, UMR-S 1172, INSERM, 59000 Lille, France; 7Medical Strategy and Innovation Department, Clariane, 75008 Paris, France; 8NeuroStim Psychiatry Practice, 75005 Paris, France

**Keywords:** Alzheimer, transcranial magnetic stimulation, precuneus

## Abstract

The current literature review aimed to evaluate the effectiveness of rTMS on the precuneus as a potential treatment for Alzheimer’s disease (AD). Although the number of studies specifically targeting the precuneus is limited, the results from this review suggest the potential benefits of this approach. Future studies should focus on exploring the long-term effects of rTMS on the precuneus in Alzheimer’s disease patients, as well as determining the optimal stimulation parameters and protocols for this population. Additionally, investigating the effects of rTMS on the precuneus in combination with other brain regions implicated in AD may provide valuable insights into the development of effective treatment for this debilitating neurodegenerative disorder.

## 1. Introduction

Alzheimer’s disease is a neurodegenerative disorder affecting over 40 million individuals worldwide. This disease is characterized by progressive cognitive decline and loss of neuroplasticity in the brain. The posterior parietal cortex plays a role in a variety of cognitive functions. These include attention, episodic retrieval, visual working memory, reasoning tasks, and the perception of pain. Additionally, it’s associated with feelings of free will [1]. Its consistent activation during episodic retrieval is not fully understood but is likely connected to attention [1]. Damage to this area results in deficits in visual working memory and the recognition of previously seen objects. The posterior parietal cortex is also activated during reasoning tasks and mathematics. The precuneus is located on the medial surface of the brain and is a part of the superior parietal lobule. The precuneus has traditionally been considered as a homogeneous structure, but recent research has revealed it to have three subdivisions: a sensorimotor anterior region, a cognitive/associative central region, and a visual posterior region. The precuneus is located between the two cerebral hemispheres, above the posterior cingulate, and forward of the cuneus (which contains the visual cortex).

It is connected to various other brain regions, including the thalamus, the claustrum, the caudate nucleus, the putamen, the zona incerta, the pretectal area, the superior colliculus, the nucleus reticularis tegmenti pontis, and the basis pontis. The precuneus plays a role in various brain functions, such as memory [2], self-awareness [3], visuospatial ability [4], executive functions, and consciousness [5]. It is involved in memory tasks like the recall of episodic memories, source memory, and familiarity judgments. It is also involved in attention, working memory, and conscious perception. Regarding visuospatial ability, the precuneus is involved in directing attention in space, motor imagery, and motor coordination. The precuneus has also been linked to mental imagery, particularly in understanding the perspectives of others and making empathetic judgments. The precuneus is also thought to be related to response inhibition and is considered “pivotal for conscious information processing”. It is one of the brain regions most deactivated during slow-wave sleep and rapid eye movement sleep [6]. It has been suggested that the precuneus is the “core node” or “hub” of the default mode network that is activated during “resting consciousness”. However, its involvement in the default network has been questioned by recent studies; one such study suggests that only the ventral precuneus plays a role. Olaf Sporns and Ed Bullmore have proposed that the precuneus functions as a central hub between the parietal and prefrontal regions, linking these clusters or modules together [7]. The precuneus has been described as a central and well-connected “small-world network” hub. A study employing inhibitory rTMS (to induce a “virtual lesion” of the precuneus) assumes that the precuneus could have a causal role in assessing one’s own memory performance (mnemonic metacognition [8]). In this study, healthy participants showing greater resting-state functional connectivity (rs-fcMRI) between the precuneus and the hippocampus, or smaller gray matter volume in the stimulated precuneal area, showed significantly higher susceptibility to the TMS effect on mnemonic metacognition (and these effects were not seen in the perceptual domain), providing strong evidence suggesting a possible network involving the precuneus and the hippocampus during the recollection of episodic details in memory.

Bonnì S et al. investigated the functional connectivity between the parietal and frontal cortex of 15 Alzheimer’s patients and 12 healthy individuals [9]. They applied conditioning stimuli over the right posterior parietal cortex (precuneus) and measured the motor evoked potentials (MEPs) from the right primary motor cortex. The results showed that, in healthy subjects, applying a TMS pulse over the parietal cortex at 90% intensity increased the excitability of the motor cortex, peaking at a 6 ms interstimulus interval (ISI). However, in Alzheimer’s patients, the same effect was only seen when TMS was applied at 110% intensity with a peak at 8 ms ISI. Additionally, treatment with cholinesterine inhibitors did not improve the strength of the connection. They stated that the effects of TMS conditioning at 110% intensity in Alzheimer’s patients correlated with cognitive abilities such as episodic memory and executive functions, suggesting that those with better cognitive performance had less impaired connectivity. These findings indicate that the functional connectivity between the parietal and frontal cortex is altered in Alzheimer’s patients and provide evidence for a disconnection-based explanation of Alzheimer’s symptoms.

Benussi A et al. aimed to evaluate the incremental diagnostic value rTMS measures in the diagnosis of Alzheimer’s disease (AD) compared to established biomarkers of amyloidosis [10]. In total, 120 patients with dementia were included in the study and scored in terms of diagnostic confidence of Alzheimer’s disease (DCAD) through a three-step assessment. The results showed that TMS measures improved the discrimination of DCAD when added to clinical and neuropsychological evaluations, with levels comparable to established biomarkers of brain amyloidosis. The classification accuracy for the gold standard diagnosis was 0.82 with just the clinical work-up. When TMS was added to the clinical work-up, the accuracy increased to 0.98. With the addition of amyloidosis markers to the clinical work-up, the accuracy further rose to 0.99. The study concludes that TMS, in addition to routine assessment in patients with dementia, has a significant impact on diagnosis and diagnostic confidence, comparable to established amyloidosis biomarkers.

In addition to its significant diagnostic value, repetitive transcranial magnetic stimulation also holds therapeutic potential and has gained increasing attention as a potential therapeutic approach for Alzheimer’s disease (AD) [11,12,13]. TMS generates a magnetic field and an electric current in targeted brain regions using a rapidly changing current delivered through a coiled wire placed above the scalp [14]. The intensity of stimulation is determined by the individual’s motor evoked potential threshold and modulates the activity of cortical neurons. Repetitive TMS (rTMS) delivers trains of pulses consistently over a set period. There are different rTMS protocols. Some use high frequency (≥5 Hz), others use low frequency (≤1 Hz). Additionally, there are various types of stimulation bursts, like theta burst stimulation (TBS). Generally, higher frequencies increase cortical excitability while lower frequencies inhibit it [15], although this may not always be the case [16]. In the motor cortex, continuous TBS has inhibitory effects, while intermittent TBS has excitatory effects. Several trials and reviews have indicated that rTMS may be beneficial for various cognitive functions in patients with Alzheimer’s disease (AD). The dorsolateral prefrontal cortex (DLPFC) is the most commonly targeted region for rTMS, as it is a key node of the central executive network [17]. Targeting the DLPFC has been shown to improve cognitive scores and is also an FDA-approved treatment for depression, which is highly comorbid with AD. rTMS targeting the DLPFC has also been shown to reduce apathy and improve cognition in AD patients. However, there are discrepancies between studies, making it difficult to determine the full effects of rTMS on AD. In Italy, Cotelli et al. conducted multiple studies on the effects of rTMS in patients with AD, including a trial with 15 patients that showed enhanced accuracy in action naming with rTMS administered to the bilateral dorsolateral prefrontal cortex (DLPFC) [18]. In a second trial with 24 adults with varying severity of AD, the researchers found that rTMS over the bilateral DLPFC improved action naming and object naming accuracy, particularly in participants with moderate-to-severe AD [19]. However, these studies only evaluated the immediate cognitive effects of a single rTMS session, and the long-term effects remain unknown. In a third trial of ten patients with AD divided into two groups, one group received high-frequency (20 Hz) rTMS over the left DLPFC for four weeks while the other received placebo rTMS for two weeks, followed by real rTMS for two weeks [20]. The authors observed that the real rTMS group had significantly higher rates of correct responses after 2 weeks of therapy, and both groups still had improved performance 8 weeks after the end of treatment. Two experiments were conducted on mild AD patients to study the effects of 1Hz rTMS on non-verbal recognition memory tasks [21]. In the first experiment, 24 patients received both real and sham rTMS over the left and right DLPFC. The results showed that real rTMS on the right DLPFC significantly improved memory performance compared to sham rTMS on the right (*p* = 0.001), but real rTMS on the left did not have a significant effect on memory performance (*p* = 0.46). The second experiment involved 14 patients who received repeated sessions of real rTMS on the right DLPFC for two weeks. The results showed a significant improvement in memory performance at the end of the treatment period and at a one-month follow-up (*p* = 0.0009 and *p* = 0.002, respectively).

Multisite stimulation using a protocol called NeuroAD [22] or rTMS-cog [23], which involves alternating stimulation of multiple brain regions. sometimes combined with cognitive training, has shown significant improvements in cognition in AD patients. However, the exact contribution of rTMS to these results is unclear, as cognitive training on its own is also beneficial. There have been studies targeting other brain areas in AD patients with rTMS. One study targeted the right inferior frontal gyrus and superior temporal gyrus to improve attention and cognitive speed [24]. Zhao J et al. investigated the effect of rTMS (20 Hz, 1 session/day, 5 days/week, 30 sessions in total) applied on the parietal (P3/P4) and posterior temporal (T5/T6) cortex on 30 Alzheimer’s disease (AD) patients, divided into mild and moderate groups [25]. The patients underwent neuropsychological tests before, immediately after, and 6 weeks after the intervention. The results showed that the rTMS group had improved ADAS-cog, MMSE, and WHO-UCLA AVLT scores compared to baseline at 6 weeks after treatment. The MoCA scores also improved in the mild AD patients receiving rTMS. Subgroup analysis showed that the effect of rTMS on memory and language in mild AD patients was superior to that in moderate AD patients. Overall, the findings suggest that rTMS improves cognitive function, memory, and language levels in AD patients, especially in the mild stage, and can be considered a promising adjuvant therapy in combination with cholinesterase inhibitors in mild AD patients. Although a previous literature review appeared to suggest that bilateral stimulation of the prefrontal cortex was the most effective [13], these studies highlight the importance of considering other brain regions for stimulation in AD treatment [26].

The role of the precuneus has received attention in the context of AD due to its involvement in various cognitive and memory processes. This idea is substantiated by recent findings indicating that rTMS applied to crucial nodes of the Default Mode Network (DMN), such as the Posterior Parietal Cortex (PPC) and precuneus, improves both short- and long-term memory functions in healthy individuals [27]. Furthermore, some previous research demonstrated that rTMS of the precuneus impacts not only the local area but also at a network level by altering the activity of the precuneus and its connections to other brain regions [28]. Some research teams, therefore, proposed that high-frequency excitatory rTMS of the precuneus could potentially enhance long-term memory in AD patients by modulating the neural activity of the PC and its connections with medial parietal and frontal regions.

In this article, we will review current research on the precuneus in AD, the potential use of rTMS as a treatment, and discuss future research directions in this field. Our goal is to provide a comprehensive overview of the current understanding of the precuneus in AD and the potential of rTMS as a therapeutic tool in order to inform future work and advance our understanding of the underlying mechanisms of this disease and the potential for effective treatments.

## 2. Method

The aim of this literature review was to examine the current research on the potential use of repetitive Transcranial Magnetic Stimulation (rTMS) on the precuneus as a treatment for Alzheimer’s disease. The following steps were taken to ensure a comprehensive and rigorous review of the literature:Database search: A comprehensive search was conducted in several electronic databases, including PubMed, Scopus, and Web of Science. The search terms used were “precuneus”, “rTMS”, and “transcranial magnetic stimulation”. The search was limited to studies published in English between the years of 2010 and 2023.Study selection: All the studies retrieved from the database search were screened for eligibility. Eligible studies were those that investigated the use of rTMS on the precuneus as a potential treatment for Alzheimer’s disease in humans. Exclusion criteria were studies that used animal models, studies that focused on other forms of brain stimulation, and studies that investigated rTMS as a treatment for other forms of dementia or cognitive decline.Data extraction: Data were extracted from the eligible studies using a standardized data extraction form. The standardized data extraction form was a systematic approach we employed to ensure consistency in collecting relevant data from the studies reviewed. This method involves a predefined template that delineates the specific variables and parameters of interest, ensuring that every researcher involved in the data collection process retrieves the same type of information, thus minimizing bias and variation in the data extraction phase. The following information was collected: authors, year of publication, study design, sample size, stimulation parameters, outcome measures, and results.Data synthesis: The extracted data were analyzed and synthesized to provide an overview of the current state of the literature on the use of rTMS on the precuneus as a potential treatment for Alzheimer’s disease.

A PRISMA diagram was created to graphically represent the study selection process and the number of studies that were included in the review. The results of the systematic review were presented in a narrative format, synthesizing the findings of the studies included in the review and highlighting the strengths and limitations of the current literature.

## 3. Results

The PRISMA diagram shows the results of the literature review, with a total of four studies being included in the analysis (Figure 1). The results were categorized into preclinical Alzheimer’s and Alzheimer’s disease.

### 3.1. Preclinical Alzheimer, Mild Cognitive Impairement, Subjective Cognitive Decline

Koch G et al. [29] conducted a double-blind, randomized, sham-controlled trial over two weeks to examine the effects of high-frequency rTMS of the precuneus on cognition in fourteen patients with early AD (seven females). Cognitive measures were derived from the Alzheimer Disease Cooperative Study Preclinical Alzheimer Cognitive Composite, which includes the Rey Auditory Verbal Learning test (evaluating long-term episodic memory), the 13 Digit Symbol Substitution Test from the Wechsler Adult Intelligence Scale–Revised (assessing response speed, sustained attention, visual-spatial skills, and set-shifting), the Mini Mental State Examination (for global cognition), and the Frontal Assessment Battery (for executive functions). TMS combined with electroencephalography (TMS-EEG) was utilized to detect alterations in brain connectivity. This study showed that rTMS of the precuneus led to a selective enhancement in episodic memory but had no effect on other cognitive domains. TMS-EEG signal analysis revealed increased neural activity in the precuneus, an amplification of brain oscillations in the beta band, and modifications to the functional connections between the precuneus and medial frontal areas within the default mode network.

Chen J et al. [30] hypothesized that rTMS targeting the precuneus in the hippocampal subiculum (HIPsub) network could potentially influence and adjust the altered HIPsub network connectivity observed in subjects with subjective cognitive decline (SCD). After identifying the potentially dysfunctional circuit, the study aimed to stimulate it using rTMS to evaluate the causal relationships in a different cohort. They hypothesized that SCD subjects will exhibit distinct changes in the patterns of HIPsub network connectivity and that these deviations in the HIPsub circuit associated with episodic memory processing could be ameliorated by rTMS targeting the precuneus in the HIPsub network of SCD subjects. They concluded that applying rTMS to the precuneus for two weeks could potentially enhance the connectivity between the hippocampal circuit (HIPc) and the left parahippocampal gyrus, as well as between the hippocampal proper (HIPp) and the left middle temporal gyrus. These improvements in connectivity could potentially lead to enhanced episodic memory performance.

### 3.2. Alzheimer’s Disease

Traikapi A et al. [31] hypothesized that the application of gamma stimulation bilaterally to the precuneus could significantly enhance the performance of patients on episodic memory tasks and might bolster the disrupted gamma activity and brain connectivity. The study’s baseline phase comprised five experimental conditions, each differentiated by its duration, ranging from one to five weeks. After each condition, patients underwent two weeks of gamma frequency transcranial magnetic stimulation (TMS). Participants were randomly assigned to these conditions, with the TMS treatment introduced at different times for each (e.g., after one week for the first participant, two weeks for the second, and so on). This design meant that participants still in the baseline phase effectively served as a control group. For instance, when the first participant received TMS, those still in their baseline phases acted as controls with no expected improvement. If TMS was the sole factor driving improvement, no behavioral changes would be expected in participants still in the baseline phase. Data were collected at different stages: pre-treatment, baseline, treatment, post-treatment, and after a three-month follow-up period. Interestingly, this stimulation also seems to have had important effects on anxiety, since the Beck anxiety inventory went from 4.75 (before treatment) to 1.5 (after), with a 68% reduction.

In a phase 2 monocentric, randomized, double-blind, sham-controlled trial, Koch et al. [32] assessed the effects of precuneus rTMS in patients with mild-to-moderate Alzheimer’s disease. Fifty participants were equally randomized to undergo either precuneus rTMS or a sham procedure. The treatment regimen spanned twenty-four weeks, commencing with daily rTMS sessions (or sham) for the initial two weeks, five times weekly. This intensive phase was succeeded by a 22-week maintenance period with weekly sessions. Notably, those subjected to the precuneus magnetic stimulation maintained their scores on the Clinical Dementia Rating Scale–Sum of Boxes, in stark contrast to the sham group, which exhibited score deterioration. Additionally, the rTMS group outperformed the sham group in secondary outcomes, such as the Alzheimer’s Disease Assessment Scale–Cognitive Subscale, Mini-Mental State Examination, and Alzheimer’s Disease Cooperative Study–Activities of Daily Living Scale. By the maintenance phase’s conclusion, the rTMS cohort showed a negligible decline in the CDR-SB score, underscoring the cognitive benefits of the treatment. This was further corroborated by secondary outcome analyses, including ADAS-COG and MMSE results.

A summary of the main findings is included in Table 1.

## 4. Discussion

### 4.1. Key Findings

This review shows that all the studies carried out on precuneus stimulation in the early stages of the disease, or even more recently in stages of confirmed disease, show a positive effect of rTMS on cognitive functions, essentially with a reduction in disease worsening. In other words, rTMS targeting the precuneus could slow the progression of the disease and preserve several cognitive functions. Precuneus-targeted rTMS appeared to be effective in reducing patients’ functional decline, as evidenced by improvement in the ADCS-ADL measure [32]). This suggests the potential utility of precuneus-targeted rTMS for treating both cognitive and functional impairments in the early stages of Alzheimer’s disease.

### 4.2. Mechanisms behind the Beneficial Effects of Precuneus Stimulation

There are several explanations as to why stimulation of the precuneus might have a beneficial effect on Alzheimer’s disease. First, regarding the effect of stimulation itself at the local level, it is possible that it increases neuronal activity in the area, as found in the study by Xu X et al. [34], who investigated the effects of theta burst stimulation (TBS), a form of transcranial magnetic stimulation, on the precuneus region of the brain. The study aimed to understand how TBS affects the local intrinsic activity in this region. To achieve this, they used two types of TBS, namely intermittent TBS (iTBS) and continuous TBS (cTBS), on 28 healthy subjects. The stimulation was applied to the left dorsolateral prefrontal cortex (DLPFC), and the local intrinsic activity of the precuneus was measured before and after the treatment. The results showed that after iTBS, significant increases in the amplitude of low-frequency fluctuation (ALFF) and fractional ALFF (fALFF) were observed in the precuneus. These measures represent the power and the relative contribution of low-frequency oscillations, which are believed to reflect spontaneous neural activity. In contrast, after cTBS, there was a significant decrease in ALFF and fALFF in the precuneus. The resting-state functional connectivity between the DLPFC and the precuneus was also affected by TBS, showing alterations depending on the type of stimulation used.

Another possible explanation is based on neuroplasticity, as found in certain studies of depressed subjects. Wang Z et al. [35] investigated the impact of rTMS on the gray matter volume in the brains of patients suffering from Major Depressive Disorder (MDD). The study involved 26 patients with their first episode of unmedicated MDD, along with 31 healthy control subjects. High-frequency rTMS treatment was carried out over 15 days, targeting the F3 point of the left dorsolateral prefrontal cortex. To observe changes in brain gray matter volume, structural magnetic resonance imaging (sMRI) data were collected before and after the treatment. Before treatment, it was found that MDD patients had significantly lower gray matter volumes in several brain regions, such as the right fusiform gyrus, left and right inferior frontal gyrus (triangular part), left inferior frontal gyrus (orbital part), left parahippocampal gyrus, left thalamus, right precuneus, right calcarine fissure, and right median cingulate gyrus, compared with the healthy controls. After rTMS treatment, a significant increase in gray matter volume of the bilateral thalamus was observed in the MDD patients, and this change could possibly be the underlying neural mechanism through which rTMS alleviates depression by enhancing neuroplasticity.

In the context of memory models based on space, the hippocampus, retrosplenial cortex, and precuneus are believed to play a part in initial memory computations. Hebscher M et al. [36] used transcranial magnetic stimulation (TMS) and magnetoencephalography (MEG) to explore the influence of the precuneus on the process of AM retrieval. When compared to vertex stimulation, stimulating the precuneus during the initial stages of memory search and construction resulted in a delay in evoked neural activity within the first 1000 milliseconds after presenting the cue. As memory elaboration progressed, the stimulation led to a reduction in sustained positivity. The study also observed a parietal late positive component during memory elaboration, the magnitude of which was tied to spatial perspective during recollection. Precuneus stimulation disrupted this association, suggesting a significant role of this region in encoding the spatial perspective during AM. These results highlight the influence of the precuneus in the early retrieval of AM, both during the memory search phase before a specific memory is accessed and during the reinstatement of the spatial context at the initial stages of memory elaboration and re-experiencing. This study, leveraging the temporal precision of MEG and the cause–effect relationship of TMS, contributes to a better understanding of the neural underpinnings of early naturalistic memory retrieval.

### 4.3. Comparative Findings between Precuneus and DLPFC Stimulation

Most studies targeting the dorsolateral prefrontal cortex (DLPFC) with rTMS have observed modest enhancements in Alzheimer’s Disease Assessment Scale–Cognitive (ADAS-Cog) scores relative to sham interventions, though often with limited participant numbers. However, this literature review underscores that all investigations focusing on the precuneus have reported a marked deterioration in the sham cohort. In contrast, the group receiving active rTMS treatment largely maintained their cognitive status, exhibiting only marginal decline. Given the neurodegenerative nature of Alzheimer’s, maintaining a patient’s cognitive state over a 6-month period is deemed a significant achievement. Nonetheless, a prevalent limitation across these studies is the absence of extended follow-up, making it challenging to ascertain the longevity of rTMS treatment benefits. This constraint is recurrent in much of the related research. Regarding safety, the collated data indicate that rTMS does not lead to lasting adverse outcomes. All documented side effects, such as headaches, scalp discomfort, neck pain or stiffness, and fatigue, were transient, resolving without intervention within a few hours post-treatment.

### 4.4. Limitations

The main limitation of this review is the small number of studies included, as well as the diversity of protocols employed (10–40 Hz, 1000–1600 pulses). However, we consider that this is an encouraging signal which, although requiring replication in future studies on larger populations, should arouse the interest of clinicians and researchers.

### 4.5. Conclusions

Overall, this literature review shows a promising protocol for stabilizing cognitive decline in patients with Alzheimer’s disease for as long as possible. It would seem logical to reproduce the rhythm proposed in some of these studies (notably that of Koch et al. [32]), where patients receive maintenance treatment once a week, but this aspect remains to be investigated.

## Figures and Tables

**Figure 1 brainsci-13-01332-f001:**
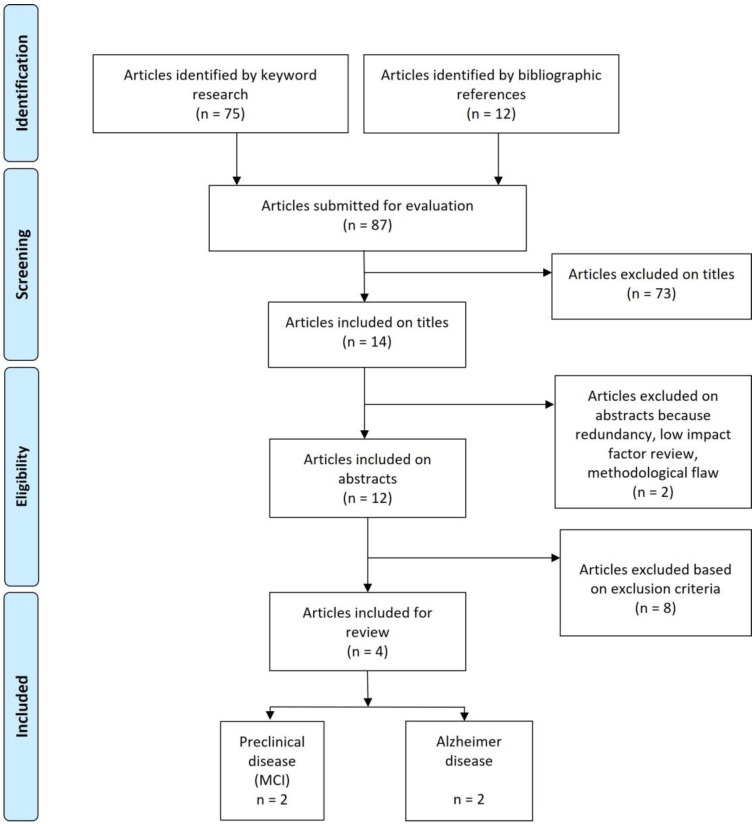
PRISMA diagram.

**Table 1 brainsci-13-01332-t001:** rTMS over the precuneus for Alzheimer disease and Subjective cognitive decline.

Reference Year	Population	Disease	Protocol Device	Main Cognitive Assessment	Outcome
Koch G et al. 2018 [29]	n = 14 7 females 7 males	Early AD	20 Hz 1600 pulses 10 sessions Over 2 weeks Adjusted MT * Neuronavigated Magstim Rapid^2^ Eight coil 70 mm	ADCS-PACC	Significant improvement at the Delayed Recall of the Rey Auditory Verbal learning Test performance after rTMS (pre vs. post, 2.42 ± 0.8 vs. 3.14 ± 0.8). No significant effects were detected after sham stimulation. No significant effects were observed on patients’ executive functions, attention, or global cognition.
Chen J et al. 2020 [30]	n = 30 22 females 8 males	SCD	10 Hz 1000 pulses 10 sessions Over 2 weeks 100% MT Pz site of the 10–20 EEG system was used to locate the precuneus Magstim Rapid^2^ Eight coil 70 mm	AVLT	Significant interactions between group (real group and sham group) and stimulation (pre-rTMS and post-rTMS) in the changes of AVLT-IR and AVLT-tot scores (*p* < 0.05). SCD subjects showed an improvement in episodic memory (AVLT) after 2 weeks of real rTMS treatment.
Traikapi A et al. 2022 [31]	n = 5 3 females 2 males	AD	40 Hz 1000 pulses 10 sessions Over 2 weeks 65–90% MT Left and right precuneus were stimulated on separate days Neuronavigated Magstim Rapid ^2^	ADAS-Cog	All patients demonstrated improvement after rTMS treatment, and average score dropped from 33 pre-treatment to 28 post-treatment. The effect was slightly more pronounced in the follow-up phase, with an average score reduction of 5.6 points (average score dropped to 27.4) compared to the pre-treatment score.
Koch et al. 2022 [32]	n = 50 26 females 24 males	mild-to-moderate AD	20 Hz 1600 pulses 10 sessions Over 2 weeks Adjusted MT * Neuronavigated Magstim Rapid ^2^ Eight coil 70 mm	CDR-SB	Patients who underwent active treatment maintained their cognitive performance, while those who received sham-rTMS demonstrated a general decline in cognitive abilities. The estimated mean change in the CDR-SB score from the start to the end of the study was −0.25 for the PC-rTMS group and −1.42 for the sham-rTMS group. The proportion of responders, defined as patients with a change in CDR-SB score of less than or equal to 1, was 68.2% in the active group and 34.7% in the sham group.

AD: Alzheimer’s Disease; ADAS-Cog: Alzheimer’s Disease Assessment Scale–Cognitive Subscale; ADCS-PACC: Alzheimer’s Disease Cooperative Study–Preclinical Alzheimer’s Cognitive Composite; * Adjusted MT: distance-adjusted MT [33]; AVLT: Auditory Verbal Learning Test; CDR-SB: Clinical Dementia Rating Scale–Sum of Boxes; SCD: Subjective Cognitive Decline.

## Data Availability

Data available on request.

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
