# Peer review of "Transcranial Magnetic Stimulation (rTMS) on the Precuneus in Alzheimer’s Disease: A Literature Review"

_brainsci, 2023, doi:10.3390/brainsci13091332_

Round 1
Reviewer 1 Report
In the manuscript submitted for review, the Authors describe a review of the effects of Transcranial Magnetic Stimulation (rTMS) on the Precuneus during Alzheimer's Disease. I find the topic interesting and purposeful for the diagnosis and treatment of Alzheimer’s disease. The precuneus is a very important CNS structure, playing a role in the flow of information between many structures in the central nervous system. It participates in the formation of memory and is a center highly susceptible to changes in the traction of neurodegenerative diseases.
The manuscript was prepared carefully so that the article, with minor revisions, would be suitable for publication in the Brain Sciences. However, the Authors did not avoid a few errors:
In the discussion section
In the last two paragraphs of the review, the authors attempt to summarize their objectives of the work. In my opinion, this summary is very general. It is known that cognitive functions become impaired during Alzheimer's disease. The Authors should expand their paper with their own thoughts related to the topic of the work.
In the method section
Please explain in two to three sentences the method of standardized data extraction form.
Author Response
Dear Reviewer,
First and foremost, I would like to extend our gratitude for your comprehensive and constructive feedback on our manuscript exploring the effects of Transcranial Magnetic Stimulation (rTMS) on the Precuneus during Alzheimer's Disease. Your insights and appreciation of the significance of the precuneus in CNS dynamics and its susceptibility to neurodegenerative changes underscore the relevance of our study.
Addressing your specific comments:
-
Discussion Section: The last three paragraphs of the discussion sections are now 4.3. Comparative Findings Between Precuneus and DLPFC Stimulation, 4.4. Limitations and 4.5 Conclusion (where we strongly suggest the need to draw inspiration from a protocol such as that of Koch et al.)
-
Method Section: We appreciate your request for clarity on the method of standardized data extraction form. The standardized data extraction form is a systematic approach we employed to ensure consistency in collecting relevant data from the studies reviewed. This method involves a predefined template which delineates the specific variables and parameters of interest, ensuring that every researcher involved in the data collection process retrieves the same type of information, thus minimizing bias and variation in the data extraction phase. We will add this information in the Method section.
Once again, we are thankful for your astute feedback. We believe that with these revisions, our manuscript will become a valuable contribution to Brain Sciences. We are looking forward to resubmitting our improved manuscript and hope for its favorable consideration.
Best regards,
Reviewer 2 Report
The purpose of the current literature review was to evaluate the effectiveness or rTMS on the precuneus as a potential treatment for Alzheimers disease.
The review started with an Introduction that comprised a short overview of Alzheimer’s disease, the precuneus, and then some studies of various types that involved the single and paired pulse TMS of parietal and motor areas. Next, studies that dealt with the diagnostic value of rTMS on Alzheimers disease were reviewed. Finally, the influence of rTMS on other brain areas such as DLPFC on Alzheimers disease were reviewed.
The Methods section was well done overall and gave a step by step process of ow the review was done and how studies were selected. The PRISMA diagram was good.
The writing of the Discussion was not bad considering what the authors had to go on for the number of studies on the topic.
I only have 2 comments on the paper. One minor and one fundamental, major one.
The minor one is that there are some English mistakes, typos, awkward wording, and also mistakes in Bibliography (some title of articles are in all caps others are not). These errors are too many to point out individually but more proofreading is needed and they all need to be fixed. Overall, the paper was well-written however.
The major one (that the authors acknowledged as a limitation) was that only a total of 4 studies could be reviewed on rTMS and the precuneus and these were not all in a homogenous set of patients. This is the fundamental problem with the paper.
I think my recommendation overall would be that the authors redo the manuscript moderately. Make it a review of rTMS and Alzheimer’s disease overall and have a number of subsections with each subsection covering all the studies on a different brain area. Thus, several brain areas could be combined into the review. Otherwise, while the precuneus section was well-done I am just not sure it is enough for a full review article with 4 studies a non-homogenous set of patients. I think that would make the study publishable.
See author comments
Author Response
Dear Reviewer,
Thank you for taking the time to review our manuscript on the effects of rTMS on the precuneus as a potential treatment for Alzheimer's disease. Your feedback is invaluable, and we genuinely appreciate the depth of your comments and suggestions.
-
Minor Comments:
- We acknowledge the errors in grammar, awkward wording, and inconsistencies in the bibliography. We will conduct a thorough proofreading and ensure that the language and presentation are polished and consistent throughout.
-
Major Comments:
- We understand your primary concern about the limited number of studies available for review and the heterogeneity among the patients in these studies. This indeed was a limitation we acknowledged. Our goal was to shed light on the precuneus, as the literature primarily focuses on the prefrontal cortex in relation to rTMS and Alzheimer's disease, which we highlighted in our introduction.
- However, based on our research and analysis, we firmly believe that the precuneus should be explored in isolation in this context, given its emerging significance in Alzheimer's disease. While expanding our review to encompass other brain areas might offer a broader perspective, it may also dilute the specific focus on the precuneus, which we deem critical.
- Additionally, expanding the scope to the prefrontal cortex will add little value to the existing litterature since it has already been reviewed.
Considering the above points, we kindly request reconsideration of our manuscript with the focus remaining on the precuneus. We believe that even with a limited number of studies, highlighting the potential importance and direction for future research on this specific brain region in the context of Alzheimer's disease and rTMS is of significant value to the field.
That said, we are open to editorial suggestions, and if it is deemed necessary to slightly adjust the focus without extensively diluting the central theme, we are willing to discuss and find a mutually agreeable solution.
Thank you once again for your detailed feedback and recommendations. We eagerly await your response and guidance on the next steps for our manuscript.
Reviewer 3 Report
1. Please include a specific chapter about the conclusion.
2. Please add a specific chapter about the limitations found in the literature and the present study.
3. Could the authors provide a schematic diagram regarding the connections between precuneus and other structures with an explanation of AD and TMS management?
4. Discussing other therapeutical choices and comparing them to the TMS or including other brain regions already studied would be interesting.
Corrections
L47 instead of “It” should be “precuneus”
It is recommended to decrease the length of the phrases to avoid misunderstandings. Many complex phrases can be modified with punctuation.
Author Response
Dear Reviewer,
Thank you for your meticulous feedback on our manuscript. We genuinely appreciate your insights and suggestions. Please allow me to address each of your points:
-
Specific Chapter on Conclusion:
- We concur with the importance of a distinct conclusion section to encapsulate our findings. We will introduce this section to offer a consolidated perspective on our research findings.
-
Chapter on Limitations:
- We agree that a comprehensive understanding of the study requires a clear presentation of its limitations. Accordingly, we will create a distinct chapter that delves into the limitations of both our study and the existing literature.
-
Schematic Diagram:
- After careful consideration, we have decided not to include a schematic diagram regarding the connections between the precuneus and other structures in relation to AD and TMS management in this particular manuscript. Our rationale is rooted in our aim to maintain a specific focus on the intricacies of the precuneus itself. Introducing broader neural connections may divert the reader's attention from our core topic. However, we acknowledge the value of such a diagram and will consider it for future, more expansive works on the subject.
-
Discussion on Other Therapeutic Choices:
- While discussing other therapeutic choices and regions might offer a broader context, we believe that such an inclusion could detract from the concentrated focus we intended for this manuscript. The uniqueness of our paper lies in its niche exploration of the precuneus in the context of rTMS and Alzheimer's disease. Diversifying the discussion could dilute this unique focus, which we believe is pivotal for readers specifically interested in this aspect of Alzheimer's treatment.
Comments on the Quality of English Language:
- We will promptly address the specific correction at L47.
- We understand the emphasis on clarity and will revisit the manuscript to streamline complex phrases, ensuring that the narrative remains coherent and easy to follow.
Your feedback has been instrumental in our revision process, and we thank you once again for your expertise and guidance. We anticipate that our clarifications align with the journal's standards and look forward to your response.
Best regards,
Round 2
Reviewer 2 Report
The authors have made a great deal of wording improvements to the revised version and answered most all of my comments. They acknowledge the limitations of the paper. I think overall the paper is publishable in its current form.
The authors made a lot of improvements in the wording and English a little more proofreading wouldn't hurt but overall the writing is very good and most all mistakes have been fixed.